# Modeling of Thermal Traces Using Fractional Order, a Discrete, Memory-Efficient Model

Krzysztof Oprzędkiewicz *,†, Maciej Rosół † and Wojciech Mitkowski †

Department of Automatic Control and Robotics, AGH University, al. A. Mickiewicza 30, 30059 Krakow, Poland; mr@agh.edu.pl (M.R.); wojciech.mitkowski@agh.edu.pl (W.M.)
* Correspondence: kop@agh.edu.pl
† These authors contributed equally to this work.

**Abstract:** In the paper, the problem of modeling of thermal traces is addressed. The proposed model allows prediction of the behaviour of a thermal mark left by a warm body on cooler ground. In the model, a fractional order (FO) approach is employed. This allows consideration of an inital function (not only a condition in one, initial time instant). The proposed model uses a scalar FO differential equation approximated with the use of Continuous Fraction Expansion (CFE) approximation. This allows maintenance of a relatively small size of the model with good accuracy in the sense of a Mean Square Error (MSE) cost function. Experimental verification confirms good accuracy of the proposed model in modeling of thermal traces.

**Keywords:** non integer order systems; 2D heat transfer; fractional order state equation; initial problem; CFE approximation; thermal camera; thermal trace





## 1. Introduction

Fractional Order (FO) models of different phenomena forming area of physics and biology have been presented by many authors (see e.g., [1–3]).

Heat conduction and transfer can also be described using fractional calculus. For example, a temperature–heat flux in a semi-infinite conductor is presented in the book [4]; the heating of a beam is presented in the paper [5]. The temperature in the room can be also modeled using a fractional order transfer function, as is presented in the paper [6].

The FO models of the one dimensional heat in the state space has been proposed in many previous papers of authors, e.g., [7–13]. Its two-dimensional generalization is presented in the paper [14]. Each proposed model has been thoroughly theoretically and experimentally verified. In each case, the FO model assures better accuracy than its IO analogue.

Models of temperature distribution obtained using thermal cameras are presented, e.g., by [6,15]. Analytical solution of the two-dimensional, integer order heat equation is proposed in the paper [16]. Numerical methods of solution of PDE-s can be found, e.g., in [17]. Fractional Fourier integral operators are analyzed u.a. by [18]. It is important to note that a significant part of known investigations deals with a steady-state temperature field with omitting its dynamics.

The analysis of the state of the art shows that discrete, state space models of images from thermal cameras employing fractional approach have not been proposed yet. Such a model should describe both dynamic and steady-state temperature distributions. It can be employed, e.g., to describe thermal processes in electronic circuits or for reconstruction of missing measurements from unavailable parts of tested surfaces. An another interesting issue is the modeling of thermal traces left a by warm body on cooler ground. This problem requires employment of a state space model due to a thermal trace which can be interpreted as an initial condition and its vanishing is modeled by the free response of a model.

The best known mathematical model of heat processes in different bodies is the Partial Differential Equation (PDE) of the parabolic type, with respect to all directions. The two-dimensional, Integer Order (IO) heat transfer equation has been considered in many papers, e.g., [19–21]. Its fractional version has been proposed in [14].

It is important to note that the models proposed by authors in papers [7–14] are dedicated to describing a system with a limited number of temperature sensors, attached in fixed places in an object. If we need to describe a whole temperature field obtained, e.g., from a thermal camera, its complexity rapidly grows. A single point, where temperature is modeled, needs to be described by a model of a minimum size of 25. For a whole modeled temperature field of size $382 \times 288$ pixels, this requires the use of an IO model of summarized size equal to 2,750,400. The use of the simplest possible FO model employing the CFE approximation requires use of a model of size 13,752,000. Such dimensions of a model exclude their use in practice to model a whole temperature field.

This motivated us to propose a much simpler model, where temperature in a single point is described by a scalar FO equation. Such a model is proposed and analyzed in this paper. Additionally, such a model allows easy investigation of a free response of a system.

This paper proposes modeling of a thermal trace with the use of the FO approach. To do it we propose a new, discrete, scalar, state space model. The discrete, free state equation is constructed using the CFE operator and approach presented in paper [9]. The proposed approach allows attainment of the accurate and relatively simple FO model. Here, it is employed to analyze a free response of a system, when the initial condition is given as an initial function. Such a method has not been presented and analyzed to date. Its novelty consists of the proposal of the fractional model which is possible to implement at a bounded platform and assuring good accuracy.

Theoretical issues presented in the paper were experimentally verified using a high speed thermal camera. The proposed approach can be useful for modeling temperature traces on various real surfaces.

In the paper the following issues are presented:

- Recalling of some elementary ideas and definitions from fractional calculus.
- The CFE-based method of solution of the FO state equation and its adaptation to solving of a scalar, initial problem.
- The use of the proposed model for modeling of thermal traces on flat wooden surfaces.
- Experimental verification of results.

## 2. Preliminaries

### 2.1. Elementary Ideas

Elementary ideas from fractional calculus can be found in many books, e.g., [4,22,23] or [24]. The deep analysis of initial problems for discrete FO systems is presented in [25]. Here only some definitions necessary to explain of main results will be recalled.

Presentation of elementary ideas is started with a definition of a fractional order, integro-differential operator. It is given, e.g., by [26].

**Definition 1.** *The elementary fractional order operator.*
*The fractional order integro-differential operator is defined as follows:*

$$
{}_aD_t^\alpha f(t) = \begin{cases} \frac{d^\alpha f(t)}{dt^\alpha} & \alpha > 0 \\ 1 & \alpha = 0 \\ \int\limits_a^t f(\tau)(d\tau)^{-\alpha} & \alpha < 0 \end{cases} \quad .
\tag{1}
$$

*where a and t denote time limits of operator calculation, $\alpha \in \mathbb{R}$ denotes the fractional order of the operation.*

Next an idea of the Gamma Euler function is recalled (see, for example [27]):

**Definition 2.** *The Gamma function*

$$\Gamma(x) = \int_0^\infty t^{x-1} e^{-t} dt. \tag{2}$$

An idea of the Mittag–Leffler function needs to be given next. It is a non-integer order generalization of exponential function $e^{\lambda t}$ and it plays crucial role in the solution of the fractional order state equation. The one-parameter Mittag–Leffler function is defined as follows:

**Definition 3.** *The one-parameter Mittag–Leffler function*

$$E_\alpha(x) = \sum_{k=0}^\infty \frac{x^k}{\Gamma(k\alpha + 1)}. \tag{3}$$

The fractional-order, integro-differential operator is expressed by different definitions. The most known are given by Grünvald and Letnikov (this is so called GL definition), Riemann and Liouville (RL definition) and Caputo (C definition). In this paper, only the C definition is applied. Its use allows expression of the linear state equation analogically, as for the IO case. The C definition is as follows [28]:

**Definition 4.** *The Caputo definition of the FO operator*

$$_0^C D_t^\alpha f(t) = \frac{1}{\Gamma(N - \alpha)} \int_0^\infty \frac{f^{(N)}(\tau)}{(t - \tau)^{\alpha + 1 - N}} d\tau. \tag{4}$$

*where $N - 1 < \alpha < N$ denotes the non-integer order of an operation and $\Gamma(..)$ is the complete Gamma function (2).*

A fractional-order linear, free, scalar state space system, employing the C definition is described below:

$$\begin{aligned} _0^C D_t^\alpha x(t) &= Ax(t), \\ y(t) &= Cx(t). \end{aligned} \tag{5}$$

where $\alpha \in (0, 2)$ denotes the fractional order of the state equation, $x(t) \in \mathbb{R}$, $y(t) \in \mathbb{R}$ are the state and output vectors, respectively, $A_{1 \times 1}$ and $C_{1 \times 1}$ are state and output matrices. For this system, only its response to initial condition will be analyzed, but this is sufficient for modeling of thermal traces.

*2.2. The CFE Approximation*

The CFE approximator allows expression of the elementary FO operator $s^\alpha$ in the form of an IIR filter containing both poles and zeros. It is faster to converge and easier to implement due to its relatively low order. It is obtained by discretization of elementary fractional order element $s^\alpha$ wit the use of the generating function in the form: $s \approx \omega(z^{-1})$. The new, discrete operator takes the following form (see e.g., [29,30]):

$$\begin{aligned} \left(\omega(z^{-1})\right)^\alpha &= g_h CFE\left\{ \left(\frac{1 - z^{-1}}{1 + az^{-1}}\right)^\alpha \right\}_{L,L} = \\ &= \frac{P_{\alpha L}(z^{-1})}{Q_{\alpha L}(z^{-1})} = g_h \frac{CFE_N(z^{-1}, \alpha)}{CFE_D(z^{-1}, \alpha)} = \\ &= g_h \frac{\sum\limits_{l=0}^L w_l z^{-l}}{\sum\limits_{l=0}^L v_l z^{-l}} \end{aligned} \tag{6}$$

In (6), $L$ is the order of approximation, $g_h$ is the coefficient depending on the sample time and type of approximation:

$$g_h = \left(\frac{1+a}{h}\right)^{\alpha}. \tag{7}$$

In (7), $h$ is the sample time and $a$ is the coefficient depending on the approximation type. For $a = 0$ and $a = 1$, we obtain the Euler and Tustin approximations, respectively. For $a \in (0,1)$, we arrive at the Al-Alaoui-based approximation, which is a linear combination of the Euler and Tustin approximants.

It is important to note that coefficient $a$ can be also employed as an additional parameter allowing fitting of the approximation to experimental data. This idea will be used during identification of the proposed model.

Coefficients $w_l$ and $v_l$ for various values of the parameter $a$ can be computed using the MATLAB function written by I. Petras and available at [31]. If the Tustin approximation is considered ($a = 1$), then $CFE_D(z^{-1}, \alpha) = CFE_N(z^{-1}, -\alpha)$ and the polynomial $CFE_D(z^{-1}, \alpha)$ can be given in the direct form (see [29]). Examples of the polynomial $CFE_D(z^{-1}, \alpha)$ for $L = 1, 3, 5$ are presented in the Table 1. The detailed analysis of various forms of CFE approximators has been given in the paper [32].

**Table 1.** Coefficients of CFE polynomials $CFE_{N,D}(z^{-1}, \alpha)$ for Tustin approximation.

| Order $L$ | $w_l$ | $v_l$ |
|:---:|:---|:---|
| $L = 1$ | $w_1 = -\alpha$<br>$w_0 = 1$ | $v_1 = \alpha$<br>$v_0 = 1$ |
| $L = 3$ | $w_3 = -\frac{\alpha}{3}$<br>$w_2 = \frac{\alpha^2}{3}$<br>$w_1 = -\alpha$<br>$w_0 = 1$ | $v_3 = \frac{\alpha}{3}$<br>$v_2 = \frac{\alpha^2}{3}$<br>$v_1 = \alpha$<br>$v_0 = 1$ |
| $L = 5$ | $w_5 = -\frac{\alpha}{5}$<br>$w_4 = \frac{\alpha^2}{5}$<br>$w_3 = -\left(\frac{\alpha}{5} + \frac{2\alpha^3}{35}\right)$<br>$w_2 = \frac{2\alpha^2}{5}$<br>$w_1 = -\alpha$<br>$w_0 = 1$ | $v_5 = \frac{\alpha}{5}$<br>$v_4 = \frac{\alpha^2}{5}$<br>$v_3 = -\left(\frac{-\alpha}{5} + \frac{-2\alpha^3}{35}\right)$<br>$v_2 = \frac{2\alpha^2}{5}$<br>$v_1 = \alpha$<br>$v_0 = 1$ |

## 3. The Considered Heat System

Figure 1 shows the simplified scheme of the heat system we deal with. This is flat surface of the wooden table with thermal camera attached vertically over table. The size of measured area is determined by the size of the sensor and the focal length of camera's lens. In the experiment, the camera OPTRIS PI 450 with lens O29 $29° \times 22°$ was used. The resolution of the camera's sensor is $382 \times 288$ pixels. The camera was attached 300 (mm) above the table, the applied lens gives a field of view of $165 \times 121$ (mm) with the size of the single pixel equal to $0.43 \times 0.42$ (mm). The diameter of the trace is equal to $\phi = 75$ (mm). Data from camera are read with the use of dedicated software OPTRIS PI Connect.

On the table, the hot cup was placed for a short time and then removed. The thermal trace left by the cup was measured by K time moments and L initial measurements ($L < K$) were used as the initial condition. The 3D temperature distributions for initial and final time instants are shown in Figure 2. The $X$ and $Y$ coordinates are given in (mm); the temperature is given in Celsius degrees. The contours of the same temperature distributions are shown in Figure 3. This figure shows also the points where time trends are drawn. To facilitate the indication of measurement points, the coordinates are given in pixels. The time trends of temperature in two exemplary points are shown in Figure 4. It is important to note

that measurements with the thermal camera can be affected by number of disturbances, e.g., by varying and unknown emissivity of the surface, light reflections and or by random ambient temperature.

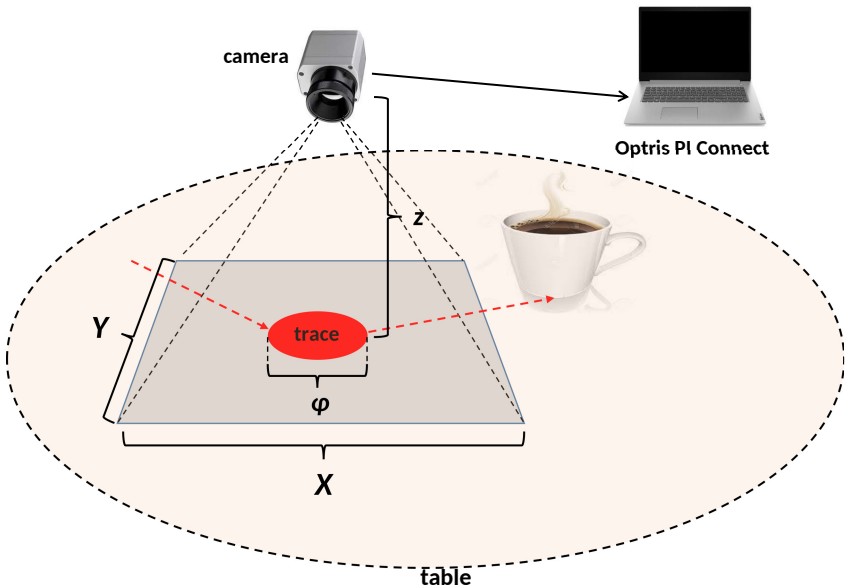

**Figure 1.** The experimental system.

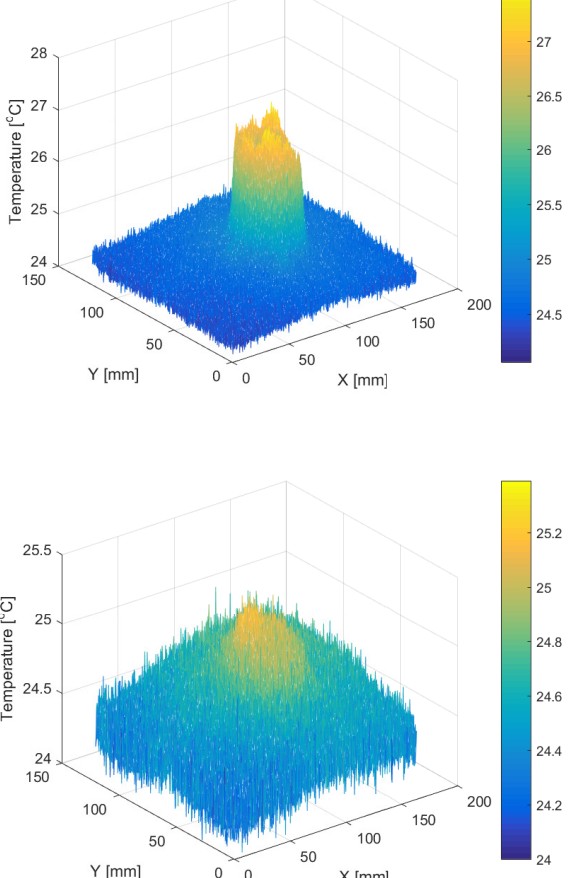

**Figure 2.** 3D diagrams of thermal traces for the initial (**top**) and final (**bottom**) time points. Dimensions *X* and *Y* in (mm).

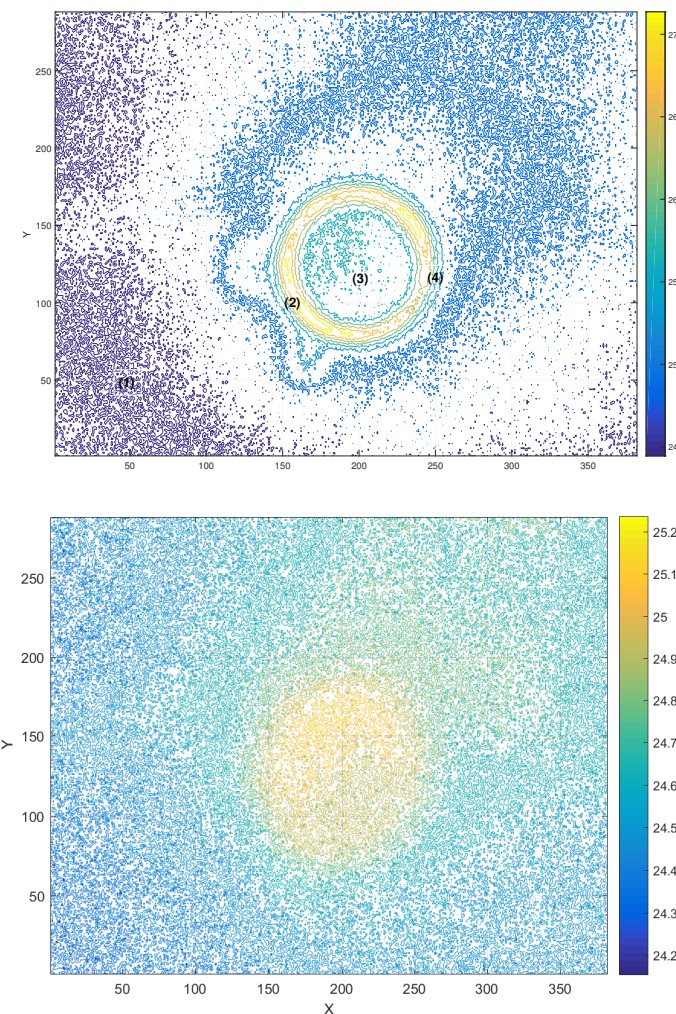

**Figure 3.** Contours of thermal traces for the initial (**top**) and final (**bottom**) time points. At the initial time point, contours are marked as the points used in experiments. Dimensions *X* and *Y* are in pixels.

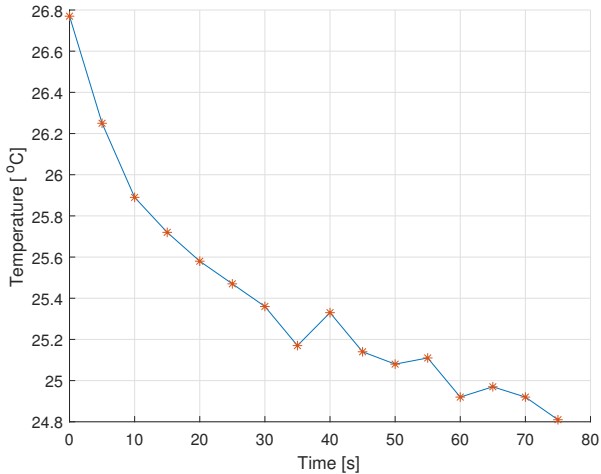

**Figure 4.** *Cont.*

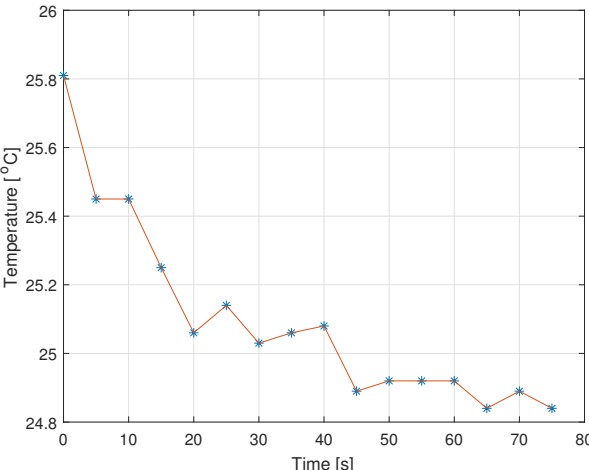

**Figure 4.** Examples of time trends of the temperature in points: (2—**top**) and (4—**bottom**).

## 4. Main Results

### 4.1. The Free, Scalar, IO State Equation

The temperature of the plate is determined by the heat transfer and dissipation. The exact model of this process, describing both time and distance dependencies, is a partial differential equation of parabolic type. However, to simplify the process, we assume that we deal with a temperature of one, the selected point of the plate, independently of others. This implies that boundary conditions do not have to be taken into account and the dynamics of the temperature can be expressed by the following, scalar, ordinary differential Equation (8):

$$
\begin{cases}
\dot{\theta}(t) = \frac{1}{T_{IO}}\theta(t), \\
y(t) = \theta(t).
\end{cases}
\tag{8}
$$

In (8), $T_{IO}$ is the time constant describing the whole dynamics of the heat transfer on the surface.

The free solution of Equation (8) is as follows:

$$
\theta(t) = \theta_0 \exp\left(-\frac{t}{T_{IO}}\right).
\tag{9}
$$

where $\theta_0$ is the initial temperature. The solution (9) will be used to compare the IO model to the discrete FO model, proposed in the next subsection.

### 4.2. The Discrete, Free, Scalar, FO State Equation Using CFE Approximation

To assure the simplicity of the model, we propose it in the form of the Fractional Order (FO), scalar Equation (5). In addition, the FO differential equation allows consideration of the initial condition in the form of a function, not as a single point. The proposed model is as follows:

$$
\begin{cases}
{}^{C}_{0}D^{\alpha}_{t}\theta(t) = \frac{1}{T_{\alpha}}\theta(t), \\
y(t) = \theta(t).
\end{cases}
\tag{10}
$$

where $T_{\alpha}$ is the time constant describing the whole dynamics of the heat transfer on the surface.

The method of the memory-effective solution of an FO state equation was proposed and analyzed by [9] and employed to solve the FO state space model of the thermal process in [13]. Using this approach the free, scalar FO (5) can be solved too. The solution takes the following form:

$$
\theta^{+}(k) = -E_0^{-1}\sum_{l=1}^{L} E_l\theta^{+}(k-l) + E_0^{-1}\sum_{l=-L}^{0}\theta_0(l).
\tag{11}
$$

where:

$$E_l = g_h w_l - \frac{v_l}{T_\alpha}, \quad l = 0, 1, \ldots, L. \tag{12}$$

In (11), the sum from $-L$ to 0 allows consideration of the initial condition given as a function defined in time instants from $L$ to 0 before the starting one.

The Formula (11) allows effectively solving of the discrete-time FO state equation using the CFE approximant and initial condition. It has the form of the difference equation with $L$ delays in the state. It can be written as an extended state equation of the dimension $L$ without delay, analogically as in the paper [33] :

$$\begin{cases} \theta_q^+(k+1) = A_q^+ \theta_q^+(k), \\ y_q^+(k) = C_q^+ \theta_q^+(k). \end{cases} \tag{13}$$

Assume that the order $L$ of CFE approximation is equal to five. This is sufficient to obtain good accuracy of approximation and allows maintenance of a reasonable size of the model. Then the state and output matrices $A_q^+$ and $C_q^+$ turn to the following form:

$$A_q^+ = \begin{bmatrix} q_1 & q_2 & q_3 & q_4 & q_5 \\ 1 & 0 & 0 & 0 & 0 \\ 0 & 1 & 0 & 0 & 0 \\ 0 & 0 & 1 & 0 & 0 \\ 0 & 0 & 0 & 1 & 0 \end{bmatrix}_{5 \times 5}. \tag{14}$$

where:

$$q_m = -\frac{g_h w_m - \frac{1}{T_\alpha} v_m}{g_h w_0 - \frac{1}{T_\alpha} v_0}, \quad m = 1, \ldots, 5. \tag{15}$$

$$C_q^+ = [1, 0, 0, 0, 0]_{1 \times 5}. \tag{16}$$

The characteristic polynomial of the extended system (13) with respect to (14) is as follows:

$$w(\lambda) = \det(\lambda I - A) = \lambda^5 - q_1 \lambda^4 + q_2 \lambda^3 - q_3 \lambda^2 - q_4 \lambda - q_5. \tag{17}$$

The discrete FO system will be practically stable if the extended system (13)–(17) is asymptotically stable. It is the IO discrete system because its "fractionality" is hidden in its parameters. This allows testing of its stability using known methods.

The initial condition for the extended system takes the following form:

$$\theta_{q0}^+ = \begin{bmatrix} \theta^+(0) \\ \theta^+(-1) \\ \theta^+(-2) \\ \theta^+(-3) \\ \theta^+(-4) \end{bmatrix}_{5 \times 1}. \tag{18}$$

where negative indices $-1, \ldots, -4$ denote time points before the initial one. This allows consideration of "a history" of temperature dynamics before the start of modeling, and it is the significant advantage over the IO model, which allows consideration of only one "point" initial condition. Using the FO mode, the initial condition can be extended to an initial function.

The summarized size of the proposed discrete, FO model for a single point of the temperature field equals five. This size is significantly smaller than a size of other fractional models or IO state space model.

## 5. Experimental Verification of Results

To verify results presented in the previous section the system shown in Figure 1 was employed. Initial and final temperature fields as well as temperature trends in selected points are shown in Figures 2 and 4.

Estimation of a performance of each model by only comparing time trends is not accurate enough. To do it, a cost function should be applied due to it giving us precise information about accuracy of a model. Additionally such a cost function can be employed for parameter identification. In this case, we use the Mean Square Error (MSE) cost function. It describes the mean difference between initial response of the plant and model at the same time points.It is an implicit function of the CFE model parameters: $\alpha$, $T_\alpha$ and $a$:

$$\text{MSE} = \frac{1}{K-L} \sum_{k=1}^{K} (y(k) - \theta_e(k))^2. \tag{19}$$

In (19), $K - L$ is the number of compared samples, $L$ is the length of initial function, $y(k)$ is the free response of the tested model. For the IO model it was computed using analytical Formula (9) in discrete time points $k = L, \ldots, K$ and initial condition $\theta_0 = \theta(hL)$. For the FO model, it was computed using MATLAB function *initial* and extended model (13). $\theta_e(k)$ is the experimental time trend measured in the same place and at the same time points $k$ using a thermal camera. The sample time during measurements was equal $h = 5[s]$, the number of all samples was equal $K = 16$, and the length of initial function was equal: $L = 5$. During the experiment, the whole temperature field was measured and next the parameters of the model were identified for selected points.

The points selected to identification are marked in the Figure 3. The parameters of the CFE model: $\alpha$, $T_\alpha$ and $a$ were obtained via numerical minimization of the cost function (19). Results of the identification are given in Tables 2 and 3. Comparison of time trends for the model vs. the experiment is shown in Figures 5 and 6.

**Table 2.** Parameters of the IO model.

| No. | $x$ | $y$ | $T_{IO}[s]$ | $MSE$ |
|-----|-----|-----|-------------|-------|
| 1 | 50 | 50 | 17,959.12 | 0.0286 |
| 2 | 165 | 100 | 1721.28 | 0.0114 |
| 3 | 200 | 125 | 3486.51 | 0.0067 |
| 4 | 250 | 125 | 6455.36 | 0.0078 |

**Table 3.** Parameters of the FO model.

| No. | $x$ | $y$ | $\alpha$ | $T_\alpha[s]$ | $a$ | $MSE$ |
|-----|-----|-----|----------|---------------|-----|-------|
| 1 | 50 | 50 | 0.4898 | 24.0387 | 1.1037 | 0.0072 |
| 2 | 165 | 100 | 0.4859 | 24.1231 | 1.1042 | 0.0102 |
| 3 | 200 | 125 | 0.4893 | 24.0492 | 1.1044 | 0.0061 |
| 4 | 250 | 125 | 0.4889 | 24.0538 | 1.1050 | 0.0102 |

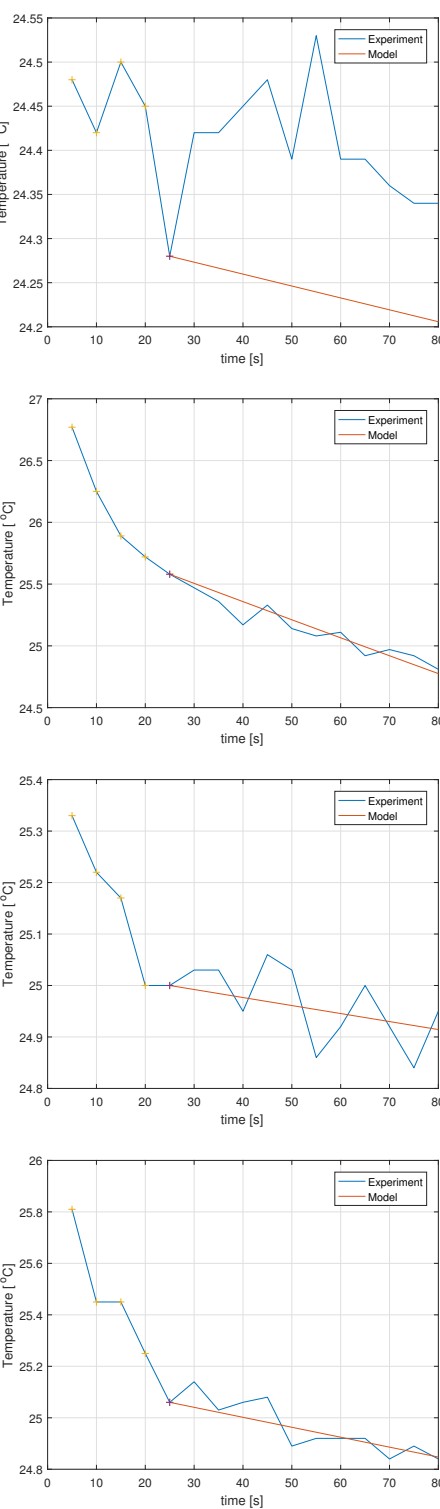

**Figure 5.** Comparison of responses of the IO model vs. the experiment for points (1—**top**)–(4—**bottom**).

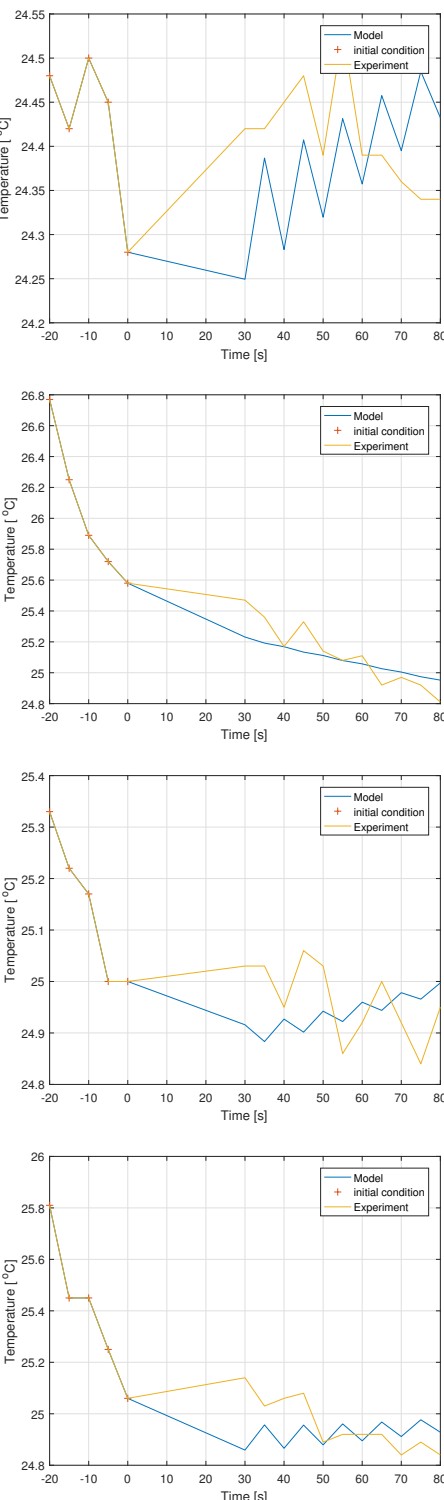

**Figure 6.** Comparison of responses of the FO model vs. the experiment for points (1—**top**)–(4—**bottom**).

## 6. Discussion of Results

The analysis of the simulation presented in the previous section allows the conclusion that:

- The FO model is more accurate than the IO model in points (1)–(3). In point (4), the IO model is more accurate.

- The sense of use of the FO model, where the initial condition is given as a function, not as a single point is well illustrated by result for point (1). At this point, the use of the initial function allows to correct prediction of the behavior of the temperature in contrast to IO model, where the prediction was incorrect.
- For the IO model, values of time constant $T_{IO}$ are significantly different for each point.
- For the FO model, the values of the parameters are very close for all the tested points of the modeled temperature field. This allows the conclusion that the proposed, simplified model correctly describes the real thermal process under consideration.

## 7. Final Conclusions

The main final conclusion from the paper is that the proposed, simplified, discrete FO model is able to accurately model the behavior of a thermal trace on typical surface. The numerical complexity of the proposed model is relatively low, which allows its use for modeling of the whole temperature field.

The proposed method allows prediction of the behavior of thermal traces using the initial condition given as a function. Such a model has not been proposed before.

The spectrum of further investigations of the presented issue is broad. The first problem is to propose a model describing the uncertainty of measurements associated to, e.g., unknown emissivity of a surface where a trace is tested. Such a model can, e.g., employ an interval approach. The next problem is the construction of the model of the whole temperature field. Furthermore, an implementation of the proposed model at multiprocessor platform, e.g., CUDA, is very interesting. An another idea is the use of image analysis methods to extract essential features of a thermal image and, next, modeling not a whole image, but only its essential parts.

**Author Contributions:** Conceptualization and supervision K.O.; Methodology W.M.; Validation M.R. All authors have read and agreed to the published version of the manuscript.

**Funding:** This research was founded by AGH grant no 16.16.120.773.

**Institutional Review Board Statement:** Not applicable.

**Informed Consent Statement:** Not applicable.

**Data Availability Statement:** Not applicable.

**Conflicts of Interest:** The authors declare no conflict of interest.

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
