# Peer review of "Modeling of Thermal Traces Using Fractional Order, a Discrete, Memory-Efficient Model"

_energies, doi:10.3390/en15062257_

Round 1

Reviewer 1 Report

The overall concept of this research is sound. The authors propose a model that allow to predict the behavior of a thermal mark left by warm body on cooler ground and use fractional order approach. It is defended that the proposed model is able to accurately model the behavior of a thermal trace on typical surface. While every thing is fine in the manuscript, the following corrections, in my opinion, should be considered by the authors;

  1. The line 116 need modification, that heading of the “3” is not appropriate. Also, I suggest to improve (line 125) the “example of 3D diagrams of thermal trace” with more explanation.
  2. The line 141 to 143 (about the significant advantage) need more explanation for the batter understanding.
  3. The literature review could be improved by include some recent relevant references on numerical and analytical solutions. For the batter attention of the readers, I suggest to includes some work on
  • Fractional HIV model using Elzaki projected differential transform method.
  • Fractional KdV equations using He’s fractional calculus
  • Analytical Solution of Linear Fractionally Damped Oscillator by Elzaki Transformed Method.
  1. Rewrite the conclusion to emphasize the work's novelty.

Reviewer 2 Report

The current paper  focuses on problem of modeling of thermal traces. . The proposed model allows to predict a behaviour of a thermal mark left by warm body on cooler ground. In the model fractional order approach is employed.

The content of the paper is well presented. However, the following minor changes are needed mandatory:

  • Why the Caputo derivative? Advantages should me mentioned.
  • Merits of fractional order approach should be explained etail.

  • The introduction section is not clear. It is very broad and not at all comprehensively focused to relevance of the manuscript. It lists a lot of other people's research work, but it does not impress on readers the importance of this work, what is not solved in this field and why is it important. Therefore, as a reader, it is difficult to draw the conclusion from them as to why this study has been carried out. The authors need to discuss the previous work instead of only mentioning that author `A' did this and author `B' did this. In conclusion, there is not any insight into the physical description of the problem studied beyond the determination of a number of parameters by means of computing software.

  • The originality/novelty is not really highlighted in Section 1 Introduction. The written paragraph focused more on what is not done, and thus will be conducted. The paragraph should elaborate more the importance, and expands/highlights the contribution of their research.
  • The literature needs to be updated/fortified with the consideration of recent bibliography.
  • Comparative study should be given for the validation of technique employed.
  • The range of physical parameters for the current analysis should be declared.
  • Some symbols are not properly addressed. Better add a Nomenclature.
  • At several places in the text the word spacing has not been taken care of.
  • Improve the physical discussion.

Reviewer 3 Report

I have read the submitted manuscript and have some comments summarized below that may be used to improve the paper.

  1. Spell out Fo and CFE in the abstract.
  2. What is the reason for the fluctuations in Fig 4? Can these be smoothed by finer grids? The source of these needs to be investigated!
  3. A comparison by the regular heat transfer model is needed! What is the result using heat transfer with appropriate boundary conditions and then compare with the fractional order model.
  4. How are the boundaries treated in the FO model? Why not regular heat transfer boundary conditions?
  5. A discussion of the accuracy of the solver is needed?
  6. A discussion on the measurement uncertainty is needed. Taking a picture with a thermal camery often prompts setting a emissivity of the surface which is often unknown (which could vary with wavelength)?

Round 2

Reviewer 2 Report

Paper can be accepted in the present form

Reviewer 3 Report

I find that all questions raised previously have been answered and that the paper now can be published.